# Transdermal Glipizide Delivery System Based on Chitosan-Coated Deformable Liposomes: Development, Ex Vivo, and In Vivo Studies

**DOI:** 10.3390/pharmaceutics14040826

**Published:** 2022-04-10

**Authors:** Mohamed M. Badran, Nadia N. Alouny, Basmah N. Aldosari, Ahlam M. Alhusaini, Amal El Sayeh Abou El Ela

**Affiliations:** 1Department of Pharmaceutics, College of Pharmacy, King Saud University, Riyadh 11495, Saudi Arabia; mbadran@ksu.edu.sa (M.M.B.); n.n.m.al@hotmail.com (N.N.A.); baldosari@ksu.edu.sa (B.N.A.); 2Department of Pharmaceutics, College of Pharmacy, Al-Azhar University, Cairo 11865, Egypt; 3Department of Pharmacology and Toxicology, College of Pharmacy, King Saud University, Riyadh 11495, Saudi Arabia; aelhusaini@ksu.edu.sa; 4Department of Pharmaceutics, College of Pharmacy, Assiut University, Assiut 71526, Egypt

**Keywords:** glipizide, deformable liposomes, chitosan, transdermal delivery, pharmacokinetics, glucose levels

## Abstract

The current study aimed to develop and evaluate a sustained-release transdermal Glipizide (GLP) film to overcome its oral administration problems. Chitosan (CS)-coated deformable liposomes (DLs) were utilized to enhance the drug transdermal delivery. The formulations were characterized in terms of particle size, zeta potential, entrapment efficiency (EE%), vesicle deformability, morphology, stability, and in vitro release. Transdermal films of chosen formulations were prepared by the solvent casting technique, and an ex vivo study throughout rat skin was also performed. Moreover, a pharmacokinetics (PK) study was carried out and blood glucose levels were estimated. All the liposomes were in the nanometer range and a high EE% was obtained from DLs compared to conventional liposomes (CL). The prepared formulations showed a high stability and the DLs exhibited a high deformability compared to CL. The in vitro release study confirmed the sustained release of GLP from both CL and DL and a more pronounced sustained release of GLP was detected after coating with CS. Moreover, GLP was shown to efficiently permeate through the rat skin from transdermal films by an ex vivo permeation test. The transdermal films showed a promising PK profile in the rat as compared with oral GLP. Most importantly, GLP-CS-DL1 demonstrated a higher hypoglycemic effect, confirming the possibility of systemic action by the local topical delivery of GLP.

## 1. Introduction

Like other drugs among the second-generation sulfonylureas, glipizide (GLP) acts by stimulating insulin secretion from pancreatic beta-cells as well as modifying the responsiveness of insulin-sensitive tissues [1]. GLP belongs to biopharmaceutical classification system (BCS) class II and has a low solubility and high permeability. One of the properties limiting the oral use of GLP is its short elimination half-life of about 2–4 h, and incompliance problems can arise [2]. In addition, severe and sometimes fatal hypoglycemia and gastric disturbances are side effects of GLP oral therapy [3]. Poorly water-soluble compounds such as GLP have solubility- and dissolution-related bioavailability problems.

Several attempts have been made to improve the bioavailability of oral GLP and hence patient compliance. A sustained-release system is one of the approaches used in several studies. It is likely that transdermal films are efficient for treating chronic disorders such as diabetes mellitus because they can successfully manage blood sugar with minimal side effects of oral delivery. This system may be particularly useful to control the blood glucose level and sustain the drug release. The administration of GLP by the transdermal route could overcome the problems associated with its oral route resulting in improved bioavailability [1,4]. To the best of our knowledge, only limited attempts have been made to facilitate the transdermal delivery of GLP. It has been reported that the solubility and permeation of GLP may hamper its application for transdermal delivery.

Due to their structural similarity with the skin constituents, biocompatibility, and biodegradability, liposomes are especially interesting options for percutaneous delivery that have been used successfully to transport drugs across the skin. There have been reports on the interaction between lipids present in the stratum corneum’s outer layers and the liposome lipid bilayer, changing the structure of the upper skin. Subash et al. concluded that glimepiride-loaded liposomes showed a successful sustained-release property [5]. Issues associated with oral gliclazide, such as frequent dosing, which leads to patients’ incompliance, have been overcome by loading in liposomes [6]. Singh et al. reported the antidiabetic efficacy of glibenclamide-loaded liposomes in alloxan-induced diabetic rats, where the level of blood glucose started to decrease from the 6th h and was maintained up to the 16th h [7]. On the other hand, because of liposomes accumulation in the stratum corneum, upper skin layers, and appendages, with minimal penetration to deeper tissues or the systemic circulation, conventional liposomes are not as advantageous as transdermal delivery systems [8].

Deformable liposome (DL) is a highly adaptable, stress-responsive, complex aggregate that can cross barriers efficiently. In contrast to conventional liposomes, whose permeation is limited to the stratum corneum’s outer layers, DLs can be transported as complete vesicles across the skin to reach the systemic circulation. This can be attributed to the fact that DL has the capability to squeeze through pores one-tenth of its diameter and thus can penetrate the stratum corneum spontaneously, hence improving the transdermal delivery of an entrapped drug [9,10]. Due to its high elasticity, DL is used as a carrier for effective transdermal drug delivery. Moreover, DLs are biocompatible, biodegradable, have a high entrapment efficiency, can protect the encapsulated drug from metabolic degradation, and can provide a sustained release [11]. The component responsible for the flexibility of DLs is the surfactant, also known as the edge activator. The addition of edge activators in a proper ratio can reduce the possibility of vesicle rupture in the skin [12]. Additionally, this allows DLs to follow the natural water gradient across the skin when applied under non-occlusive conditions. Sodium cholate, Span 80, and Tween 80 are examples of some edge activators that have been used [13]. The generation of an osmotic gradient governs the mechanism of DL penetration. The difference in water content between the stratum corneum and epidermis causes the occurrence of an osmotic gradient, which acts as a driving force that results in the penetration of DLs through the skin. Another use of DLs is as skin penetration enhancers that can disrupt the intercellular lipids, leading to an enlargement in the size of skin pores [14].

The surface modification of liposomes is one of the most promising ways to improve their stability. Some improvements have been attained in the chemical and physical stability of polymer-coated liposomes prepared with poloxamer, chitosan (CS), carboxymethyl chitosan, and dextran derivatives [15]. Coating the surface of liposomes vesicles with a polymer such as CS has been found to improve their stability and minimize the leakage of the entrapped drug [16]. Chitosan has been used to coat the surface of negatively charged liposomes due to electrostatic interactions between the negatively charged phospholipids and the positive charges of chitosan’s primary amino groups. CS can improve the stability of liposomes by creating a repulsive interaction—i.e., long-range mutual repulsion between the adjacent bilayers to prevent the close approach of liposome vesicles [17].

Moreover, the use of CS can increase the efficacy of the loaded drug because it ensures a sustained release and improves the cellular uptake of the liposomes by cells owing to the chitosan positive charge. In addition, CS mucoadhesive properties will prolong the residence times at the site of drug absorption due to the increased contact of the carrier with the absorbing site. Previous studies have confirmed the important role of CS in the transdermal delivery system. Park SN. et al. observed that CS-coated liposomes had superior permeation compared to the uncoated sample [17]. Yadav AV and Urade MN prepared transdermal patches of lornoxicam using chitosan as a release-controlling polymer, increasing its permeability and perhaps enhancing the bioavailability of the drug [18]. Ali HS and Hanafy AF found that the transdermal delivery of glibenclamide from a biodegradable chitosan patch provided better maintenance of the drug’s glucose levels in the blood for a prolonged period. This was evidenced by a lower incidence of hypoglycemia associated with oral therapy [19].

The interest of this work includes the association of the transdermal delivery of GLP and CS-DLs as novel carriers of delivering GLP. This approach could be an effective and noninvasive sustained-release formulation to achieve the hypoglycemic effect. Therefore, this study was designed to develop GLP-loaded DL containing two types of edge activators (Tween 80 and sodium deoxycholate) and coated DL with chitosan to obtain chitosomes. The prepared formulations were assessed in terms of particle size, zeta potential, polydispersity index, GLP entrapped efficiency, vesicle morphology, and in vitro GLP release, while stability and deformability studies were also carried out. Transdermal films of GLP were prepared and evaluated. Pharmacokinetics and pharmacodynamics studies were performed following transdermal application to adult male Wistar albino rats after the induction of diabetes using the Streptozotocin model.

## 2. Materials and Methods

### 2.1. Materials

Glipizide was purchased from A.M.S.A. Anonima Materie Sintetiche and Afini (S.P.A., Como, Italy). Lipoid S100 (phosphatidylcholine (PC), soybean lecithin, >94% PC) was obtained from Lipoid GmbH (Ludwigshafen, Germany). Cholesterol and low-molecular-weight chitosan (viscosity 20,000 cps), degree of deacetylation (DD of 92%) were purchased from Sigma-Aldrich Chemical Co. Ltd. (St. Louis, MO, USA). Sodium deoxycholate, methanol, and acetonitrile (HPLC grade) were obtained from Fisher Scientific Co., (Hampton, NH, USA). Tween 80 was obtained from BDH Organics (England, UK). Hydroxypropyl methylcellulose (HPMC) low viscosity (viscosity of ~3 cP for 2% aqueous solution), MWT ~20,000 Da; (HPMC LV grade E3) was purchased from Dow Chemical Co. (Midland, MI, USA). Propylene glycol (PG) was obtained from WINLAB Laboratory chemical reagents (Leicestershire, UK). Streptozotocin and sodium carboxymethyl cellulose (average molecular weight of ~90,000) were purchased from Sigma-Aldrich (St. Louis, MO, USA). All the other chemicals were of analytical grade.

### 2.2. Methods

#### 2.2.1. Preparation of GLP-Loaded Deformable Liposomes and CS-Coated Liposomes

Deformable liposomes (DL) were prepared using different molar ratios of lipoid S100 (PC), cholesterol, Tween 80, and sodium deoxycholate, as shown in Table 1. Moreover, conventional liposomes (CL) were obtained without the addition of surfactants. The amount of GLP was kept constant in all formulations (0.5% *w*/*v*). The thin-film hydration method was adapted to prepare the DL and CL, with slight modifications [20,21]. Briefly, in a clean dry round-bottom flask, PC, cholesterol, surfactants, and GLP were dissolved in a suitable volume of a mixture of chloroform:methanol (2:1) to obtain a clear solution. Subsequently, the organic solvent was eliminated by evaporation through a rotary evaporator (Buchi, Switzerland) at 45 °C for 30 min under reduced pressure (258 mbar) and rotated at 200 rpm followed by purging with nitrogen gas.

The resultant dry film was hydrated with an appropriate volume of pre-warmed distilled water at 45 °C and vortexed for 30 min to form multilamellar vesicles (MLVs) of liposomes. The formed MLVs were then subjected to a probe sonication (Badnelin, Germany) for 10 min to achieve a vesicle with a nano-size range. Light centrifugation was employed at 5000 rpm for 5 min to remove the metals shaded from the probe sonicator.

To obtain CS-coated liposomes, chitosan (CS) stock solution (1.0% *w*/*v*) was obtained by dissolving CS in 0.5% (*v*/*v*) acetic acid solution (pH = 5.5; adjusted by 2 M sodium hydroxide) under magnetic stirring at room temperature for 5 h and filtered using a vacuum pump (GAST Manufacturing, Inc. Mich. Auburn, WA, USA). The CS-coated liposomes were prepared by the dropwise addition of an equal volume of liposomes to specific concentrations of CS solutions (0.2% *w*/*v*) under magnetic stirring at room temperature for 30 min. Several chitosomes were obtained containing 0.1% *w*/*v* CS, as shown in Table 1. The amount of GLP was kept constant in all formulations (0.5% *w*/*v*). The obtained formulations were stored in a refrigerator at 4 °C for further studies.

#### 2.2.2. Physicochemical Characterization of the Formulations

##### Vesicle Size Distribution and Zeta Potential Measurements

The particle size and zeta potential are well-known key parameters that can affect the performance of the CS-DLs to overcome skin barriers. The vesicle size and polydispersity index (PDI) of the formulations were measured by dynamic light scattering using a Zetasizer Nano ZS (Malvern Instruments, Ltd., Worcestershire, UK). A clean cuvette was filled with a 1:100 solution of the formulations to filter deionized water, equilibrated at 25 °C, and then subjected to vortexing for 3 min. Zeta potential was determined by the Zetasizer Nano ZS (Malvern Instruments, Ltd., Worcestershire, UK) based on electrophoretic mobility after proper dilutions were performed. All the results are presented as the average of triplicate measurements.

##### Determination of the Entrapment Efficiency (EE)

Entrapment efficiency (EE) is another parameter to investigate the efficiency of the obtained formulations. The EE% was established by the indirect method using the ultracentrifugation technique. Firstly, an accurate volume of each formulation was dissolved in methanol (1:10), and this represented the total amount of drug (W_total_). Secondly, a certain volume of each formulation was exposed to ultracentrifugation (Optima^TM^ Max-E, Ultra Centrifuge Beckman Coulter, Pasadena, CA, USA) at 4 °C for 45 min. After that, the resultant supernatant was dissolved in methanol; this represented an un-entrapped (W_free_) fraction. All the samples were filtered using a 0.45 μm syringe filter (Millipore Milex-HN, Merck, Germany) before being analyzed by a reversed-phase HPLC method with slight modification [22]. Then, the EE% was calculated according to the following equation [23,24]:EE (%) = [(W_total_ − W_free_) /W_total_] × 100

##### The Vesicle Deformability Study

The deformability of the liposomes was assessed using the extrusion method with slight modifications [25]. Briefly, CL and DL (500 µL) were extruded using an extruder (Avestin Inc., Vancouver, Canada) through double polycarbonate membranes with a 100 nm pore size (Nucleopore^®^, Whatman, GE Healthcare Bio-Sciences, Pittsburgh, PA, USA). The size of the vesicles was determined before and after the extrusion by dynamic light scattering using a Zetasizer Nano ZS. The deformability index (DI) of the obtained vesicles was qualified according to the following equation:DI (%) = [(Size before extrusion − Size after extrusion)/Size before extrusion] × 100

##### Morphological Characterization

The morphology of the selected CS-coated DL formulations was visualized using transmission electron microscopy (TEM). They were stained with 2% urany1 acetate solution, and the excess was removed using filter paper. The samples were allowed to air-dry in a dust-free environment and then observed by TEM (JEM-1011, JEOL, Tokyo, Japan) at 60 kV.

#### 2.2.3. In Vitro Release Study

An in vitro drug release study was achieved using Franz diffusion cells (Franz diffusion cells (Logan Instrument Corp., Somerset, NJ, USA). Initially, the saturated solubility of GLP in the receptor fluid (0.5% *v*/*v* Tween 80 in PBS; pH 7.4) was determined to ensure the sink condition during the experiment. An artificial cellulose membrane (MwT cut-off 8000–14,000; Livingstone, NSW, Australia) was mounted between the diffusion cells’ donor and receptor compartments. The receptor medium (12.5 mL) was composed of 0.5% (*v*/*v*) Tween 80 in PBS (pH 7.4) and adjusted at 37 °C. The formulations were added to the donor compartment, then at fixed time intervals (0.25, 0.5, 1, 1.5, 2, 4, 6, 8, and 24 h) an aliquot of 1 mL was withdrawn from the receptor media and replenished with an equal volume of fresh receptor medium [26]. All samples were filtered using a 0.45 μm syringe filter before being analyzed by reversed-phase HPLC. The cumulative amounts of GLP released were calculated as a function of time. Moreover, the in vitro drug release study was carried out from GLP suspension (0.5% *w*/*v*) in water (GLP control) for comparison. All experiments were performed in triplicate. A kinetic analysis of the in vitro release data was performed after fitting the obtained data to different kinetic models: zero-order, first-order, Higuchi, and Korsmeyer–Peppas.

#### 2.2.4. Physical Stability of the Liposomes and CS-Coated Liposomes

The selected samples were kept at 4 °C for a short-term stability study. Aliquots of the stored samples were taken after 7, 14, 21, and 28-day time intervals. These samples were investigated for any possible aggregation by macroscopic observation. Furthermore, alteration in the vesicle size and vesicle size distribution was detected by a Zetasizer Nano ZS. All experiments were performed in triplicate.

#### 2.2.5. Preparation of GLP-Loaded CL, DL, and CS-Coated Liposomes Transdermal Films

All the prepared formulations were selected to be incorporated into transdermal films. The films were prepared using the solvent casting technique [27,28]. HPMC (1% *w*/*v*) and PG (1% *w*/*v*) were dissolved in distilled water using a magnetic stirrer to prepare the casting solution. The casting solution was subjected to sonication for 1 h to remove air bubbles. Then, it was left overnight in the refrigerator at 4 °C to attain a clear solution. An accurate volume of each liposome formulation containing GLP equivalent to 0.5 mg/cm² was added to an exact volume of casting solution to provide a total volume equal to 1 mL/cm² under magnetic stirring for 15 min at room temperature. Sonication was performed for 30 min to remove the air bubbles produced by stirring. The solution was poured into a Teflon plate with a 6.3 cm diameter (about 31 cm² area) and kept in an oven at 40 °C until the complete evaporation of the water. After drying, the film was carefully peeled off from the Teflon plate and then cut into pieces of 1.6 cm diameter. The obtained films were stored in aluminum foil bags and placed in a desiccator for further experiments. For pure GLP transdermal film, the same amount of GLP as used in GLP DL and chitosomes was dispersed in the polymer solution. The same procedure as mentioned above was followed.

#### 2.2.6. Evaluation of GLP-Loaded CL, DL, and CS-Coated Liposomes Transdermal Films

The film’s thickness was measured using a screw gauge micrometer (Mitutoyo Company, Aurora, IL, USA) at five different points on the film and the average values were calculated. The weight variation of the five films (2.01 cm²) was determined separately using a digital balance (Mettler Toledo, Columbus, OH, USA) and the weight average was calculated [29]. Moisture uptake can impact the mechanical strength and drug release of a transdermal therapeutic system. Three films from each batch were weighed separately and the average weight was calculated. This weight was considered to be an initial weight. The films were then kept in desiccators that contained a saturated solution of sodium chloride (357 gm in 1 L of distilled water) for 24 h at room temperature. After 24 h, the films were reweighed and considered as final weights, and then the moisture uptake% was determined from the following equation [18]:Moisture uptake (%) = [(Final weight − Initial weight)/Final weight] × 100

The drug content of the prepared films was evaluated. Films were cut into small pieces (about 2.5 cm²) and dissolved in a 100 mL mixture composed of phosphate-buffered saline pH 7.4 and methanol in a ratio of 50:50. The solution was shaken continuously in a shaking water bath (JULABO Inc. PA, USA) adjusted to 100 rpm at 25 °C for 24 h. Furthermore, to confirm the complete extraction of the drug from the film, the solution was sonicated for 1 h under stirring for 5 min. The solution was filtered using a 0.45 μm syringe filter (Millipore Milex-HN, Merck, Germany) and analyzed using reversed-phase HPLC [30]. The folding endurance was determined manually for the obtained films. The folding endurance of the films was measured by repeatedly folding a film measuring 2 cm × 2 cm in size at the same place until it broke. The number of times the film could be folded at the same place without breaking gave the value of folding endurance [30].

#### 2.2.7. Ex Vivo Skin Permeation Study

The ex vivo experimental protocol was approved by the Research Centre Ethics of King Saud University, College of Pharmacy, Riyadh, Saudi Arabia (Ref. No.: KSU-SE-19-73). Male Wistar rats weighing 200–250 g were anesthetized and the abdominal skin’s hair was removed. Full-thickness abdominal skin was excised and the subcutaneous fat was carefully removed. The skin was washed with PBS and stored at −20 °C till further use [31,32]. An ex vivo skin permeation study was performed utilizing identical circumstances as the in vitro release study described above. Rat skin pieces were equilibrated in PBS pH 7.4 for 30 min before being mounted on Franz diffusion cells. A circular section of the prepared transdermal film with a radius equal to 0.8 cm and an effective diffusion area of 2.01 cm² was placed in the donor chamber on the skin’s stratum corneum side. The receptor compartment consisted of PBS at pH 7.4 and 0.5% (*v*/*v*) Tween 80, and it was adjusted to 37 °C. The amounts of the GLP that permeated at several time points (0.25, 0.5, 1.0, 1.5, 2.0, 4.0, 6.0, 8.0, and 24.0 h) were analyzed by HPLC. All samples were analyzed in triplicate and the permeation profiles were constructed by plotting the cumulative amount of GLP permeation versus time.

#### 2.2.8. Permeation Parameters

The cumulative amount of GLP that permeated through the skin membranes was plotted as a function of time. The slope and intercept of the linear portion of the plot were derived by regression. GLP fluxes (J, μg/cm^2^/h) through the rat skin were calculated from the slope of the linear portion of a plot of the cumulative amount that permeated through the rat skin per unit surface area versus time; the extrapolation of this line intercepted the x-axis at a time equal to the lag time (T_L_). The permeability coefficients (Kp, cm/h) were obtained by dividing J by the initial drug concentration (C_0_) in the donor compartment.

#### 2.2.9. Scanning Electron Microscopy (SEM)

The morphology of the transdermal films of the selected formulations was visualized by scanning electron microscopy (JEOL-JSM-840A, Tokyo, Japan). The films were sputter-coated with gold before scanning.

#### 2.2.10. In Vivo Study

Among the transdermal films that were prepared earlier, GLP-CS-DL1 and GLP-CS-DL3 were selected for further in vivo experiments based on their high cumulative amounts of drug permeation at the end of 24 h.

##### Experimental Animals

Adult male Wistar albino rats weighing 200–250 g were used, and the in vivo experimental protocol was approved by the Research Centre Ethics of King Saud University, College of Pharmacy, Riyadh, Saudi Arabia (Ref. No.: KSU-SE-19-73). The animals were acclimated to the laboratory conditions for at least 1 week before the start of the experiments. The animals were kept in standardized conditions (22 ± 5 °C, 55 ± 5% humidity, and 12 h light/dark cycle). They were housed in polypropylene cages at four per cage with free access to water and chow diet ad libitum.

##### Pharmacokinetics Evaluation

Twenty overnight-fasted rats whose hair has been removed earlier were divided into four groups of five rats each. Group 1 was given oral GLP suspension (5 mg/kg) in 2.5% *w/v* of sodium carboxymethyl cellulose. The animals were treated with GLP-CS-DL1 transdermal film (group 2) and GLP-CS-DL3 transdermal film (group 3) using an equivalent of 5 mg/kg of GLP. Group 4 was treated with untreated GLP transdermal film (GLP-TDF) (equivalent to 5 mg/kg of GLP). Before applying the transdermal films, the skin was gently wiped with warm water and additionally secured by medical adhesive tape. The blood samples were withdrawn at different time intervals (0, 0.5, 1, 2, 4, 6, 8, 24, and 48 h) through the tail vein using heparinized tubes. The plasma was separated by centrifugation and stored at −20 °C until further examination. A rat plasma sample of 0.1 mL and 0.1 mL of 0.1 N HCl was vortexed for 3 min, then 3 mL of acetonitrile was added for the precipitation of the plasma proteins. The mixture was shaken using a vortexer for 5 min followed by centrifugation for 10 min at 6000 rpm, and the precipitates were removed by a 0.22 μm syringe filter. The organic phase was evaporated under a nitrogen environment and the mobile phase was added to this residue followed by vortex mixing. Pharmacokinetic studies of GLP were performed using a reversed-phase HPLC method.

##### Pharmacokinetics (PK) Analysis

The results obtained for the plasma concentration of GLP were presented as the mean ± S.D. A PK analysis was performed with the aid of Excel 2010 using the PK Solver Add-In software. The elimination rate constant (K) was estimated from the linear regression analysis of the terminal portion from the log-linear plasma concentration–time profile. The elimination half-life (t_1/2_) was calculated from the terminal elimination rate constant using the formula t_1/2_ = 0.693/K. The peak plasma concentration (C_max_) and its corresponding time (T_max_) were read directly from the plasma drug concentration–time profile. The area under the curve for each drug concentration–time curve from 0 to 48 h AUC_0−48_ was calculated using the trapezoidal rule.

##### Induction of Hyperglycemia

The streptozotocin model was used to induce hyperglycemia in the diabetic rats (*n* = 5). Streptozotocin (55 mg/kg) solution was freshly prepared in 0.1 M citrate buffer at pH 4.5. The rats were fasted overnight before the induction of diabetes. The prepared streptozotocin solution was administered via intraperitoneal (i.p.) injection. A 5% (*w*/*v*) glucose solution was given to animals to overcome drug-induced hypoglycemia for the next 24 h. After 24 h, the fasting blood glucose level was measured to confirm type 2 diabetes. The blood glucose levels were determined using an Accutrend Alpha Glucometer (Roche Diagnostics, Mannheim, Germany). The degree of hyperglycemia was evaluated, and rats with blood glucose levels > 250 mg/dL were chosen to complete the study [22].

#### 2.2.11. Statistical Data Analysis

Data analysis was carried out using Microsoft Excel, version 2010, and the Origin software, version 8. Statistical differences between groups were analyzed by one-way analysis of variance (ANOVA). The values at *p* < 0.05 were chosen as statistically significant. Results are expressed as the mean ± standard deviation.

## 3. Results and Discussion

Liposomes are one of the most successful carriers used for transdermal delivery, because of their interesting properties such as biocompatibility and FDA approval for human use. In addition to the drug protection capability of PCL from degradation, PCL NPs have the ability to offer sustained drug release. When CS is used to coat liposomes, significant structures are attained with improved drug skin permeability. Moreover, the surface modification of liposomes by CS might allow the nanocarriers to bind to the cellular membrane. The cost-effectiveness considers the treatment with liposomal formulations more expensive, more effective, and safer than conventional formulations, with the highest cost concerning the lipid used. Furthermore, the cost-effectiveness occurs from being able to take a lower dose for the same outcome. This recommendation should be followed to increase the efficacy of the delivery system while confirming its safety once inside the body, as well as the cost-effectiveness of nanocarriers in medicinal applications.

### 3.1. Physicochemical Characterization of GLP-Loaded CLs, DL, and CS-Coated CLs and DLs

#### 3.1.1. Vesicle Size Distribution and Zeta Potential Measurements

All the liposomes investigated were in the nanometer range, as shown in Table 2. In the current study, Tween 80 and sodium deoxycholate were utilized as edge activators in the DL preparation. The average vesicle sizes were 109.57 ± 5.94, 154.92 ± 15.82, 144.16 ± 5.20, and 247.8 ± 5.75 nm for GLP-CL, GLP-DL1, GLP-DL2, and GLP-DL3, respectively. The vesicle size of the DLs was increased by the existence of a surfactant, which might be due to a reduction in the phospholipid concentration in these formulations [33]. With respect to the type of surfactant formulations prepared with Tween 80, DL1 showed a larger vesicle size than sodium deoxycholate (DL2). This could be attributed to the higher negative zeta potential (−29.80 ± 3.36), leading to repulsion between the bilayers and thus increasing the size of Tween-containing vesicles. Moreover, the addition of sodium deoxycholate to Tween 80 was observed to increase the size of GLP-DL3. These results were consistent with those of other studies that used the same types of surfactants, including Tween 80 and sodium deoxycholate [34,35]. It was concluded that vesicles up to 300 nm can release their drug in the skin’s deep layers [36].

Moreover, coating liposomes with CS is one of the approaches used to enhance their stability and increase the loaded drug’s efficacy, since it assures a sustained release [15]. This behavior is due to electrostatic interactions between the negatively charged phospholipids and the positive charges of a primary amino group of CS [26]. The vesicle size of liposomes was increased after coating with CS-produced formulations in the nanometer range. These significant increases in liposomal size were ascribed to the adhesion of the CS to the liposome surface via strong interactions, thereby forming a layer on the surface.

PDI is an indicator of particle homogeneity, and its value ranges from 0.0 to 1.0. As the value is close to zero, the vesicle’s homogeneity becomes higher [37]. From Table 2, it can be seen that the liposomes and CS-coated liposomes had a PDI of less than 0.5, meaning a moderate homogenous size and distribution.

Zeta potential values give an indication of the surface electrical charge of the vesicles, which is a particularly significant parameter that affects liposome behavior [22]. A high zeta potential value is expected to indicate a more stable system due to a strong electrostatic repulsion that can prevent liposomes’ aggregation [22]. The obtained uncoated liposomes (CL, DL) had a negative zeta potential ranging from −25.71 ± 5.90 to −29.80 ± 3.36 for GLP-CL and GLP-DL1, respectively. The magnitude of this value was sufficiently high to provide a good stability for the vesicles, as shown in Table 2. Moreover, Tween 80 showed a positive influence and sodium deoxycholate has a less negative impact on the zeta potential values but together showed a positive impact. These findings were in agreement with the results of other studies that used surfactants, including Tween 80 and sodium deoxycholate, in the preparation of DLs [20,23].

The zeta potential of the CS-coated liposomes was measured and is displayed in Table 2. All the formulations were observed to have positive zeta potential values. The change of the surface charge from negative to positive confirmed the interaction between the liposomes and CS. Furthermore, positively charged liposomes were found to provide better skin permeation, which could be valuable in the transdermal application of GLP [16].

#### 3.1.2. Determination of the Entrapment Efficiency (EE)

The EE% results (Table 2) showed that GLP was entrapped with diverse values of 79.48 ± 0.55% 98.65 ± 0.01%, 87.32 ± 0.46%, and 93.25 ± 0.12% for GLP-CL, GLP-DL1, GLP-DL2, and GLP-DL3, respectively, and a higher EE% was obtained from DLs than CLs. This behavior is due to the surfactant intercalated into the bilayer of DL, which altered the packing density and fluidity of the vesicles’ polar–nonpolar interface, resulting in the high permeability of GLP [20]. The different EE% of the DL formulations depends on the type of surfactant used. The DL consisting of Tween 80 (GLP-DL1 and GLP-DL3) had a higher EE% than the one that contained sodium deoxycholate (GLP-DL2). These findings were in agreement with other studies that found that Tween 80 produced a higher EE% [20]. Depending on hydrophilic:lipophilic balance (HLB) values, the affinity for lipids was expected to be higher in Tween 80 with an HLB value of 15 than in sodium deoxycholate with an HLB of 16. As the HLB decreased, the lipophilicity increased, resulting in the improved solubility of a lipophilic drug in the bilayer [20].

The EE% of GLP in different CS-coated liposomes formulations ranged from 51.90 ± 1.25% (GLP-CS-DL2) to 88.59 ± 0.37% (GLP-CS-DL1). These results demonstrated that the CS-coated liposomes had a decreased EE% compared to uncoated liposomes. The reduction in EE% observed in the CS-coated liposomes might be explained by the fact that CS is expected to be both vesicle entrapped and surface available [38]. Moreover, CS is a positively charged polymer; the EE% reduction could thus be attributed to the interactions between CS and the polar head groups (negatively charged) of the phospholipid bilayers. A higher EE% value was obtained in the formulation that contained Tween 80 as an edge activator (88.59 ± 0.37%; GLP-CS-DL1). Tween 80 is a non-ionic surfactant; it is less hydrophilic (HLB 15), providing more interaction with the hydrophobic GLP [39]. Compared with sodium deoxycholate, lower EE% values were obtained, which may be attributed to an increase in the hydrophilicity of the vesicles, resulting in more CS being encapsulated instead of the drug. Therefore, Tween 80 showed a positive effect and sodium deoxycholate had a negative impact on EE% and together surfactants showed a positive impact. The existence of CS on the surface of the liposomes had a high negative impact on EE%.

#### 3.1.3. Deformability of Liposomes

The deformability principle is based on the ability of the vesicles of liposomes to maintain their size after extrusion through a polycarbonate membrane, which has a certain pore size. The change in the vesicle size was expressed in terms of the deformability index (DI). The DI is defined as the degree to which the liposomes deform following extrusion and do not recover to their original size. The greater the DI, the less deformable the liposomes that were unable to regain their previously large size [40]. The CL showed the highest DI, 38.7 ± 0.89%, which means it had the highest difference between the average vesicle size before and after extrusion. In contrast, DL1, DL2, and DL3 showed lower DIs of 7.0 ± 0.67%, 13.8 ± 3.27%, and 8.4 ± 3.24%, respectively, and these results are in agreement with those of the previous study [41]. The results confirmed the ability of these vesicles to be deformed due to the presence of edge activators causing the liposomes to change their shape under stress without rupture. It is well known that edge activators can combine with the bilayer phospholipid membrane of vesicles; this action may perturb the acyl chains and increase the deformability of the vesicle membrane [42]. DLs have the ability to enhance the delivery of the drugs through the intact skin membrane due to their deformable characteristics that overcome the problems of CLs. The main mechanism for this behavior is based on the distribution of water content, which is low at the outer skin layers. Therefore, DLs transport the drug deeply across the intact skin under the effect of a water gradient [41].

#### 3.1.4. Morphology of Liposomes

The transmission electron microscope (TEM) is a significant description method for the size and shape of liposomes. TEM images presented in Figure 1 show that GLP-CS-DL1 and GLP-CS-DL3 had a good dispersion and oval- and irregular-shaped vesicles, which might be due to their membranes’ deformable properties.

### 3.2. In Vitro Release Study

It was reported that positive drug therapy can be controlled by the residence time of the nanocarriers in vivo. Therefore, an in vitro release study may imitate a vesicle’s behavior inside the body based on the residence time. The GLP release profiles from liposomes and chitosomes were evaluated using the Franz diffusion technique. The in vitro release of GLP through CS-coated liposomes was found to be slower than that through liposomes, which was thought to be due to the presence of CS. The release behavior of GLP from the CL-, DL-, and CS-coated CL/DL as a function of time is illustrated in Figure 2, demonstrating the influence of the liposome composition on the GLP release.

By using a GLP suspension (control), 5.55 ± 1.64% of the drug released after 2 h was detected. However, a significant increase was observed in the drug release up to 13.01 ± 1.59% after 24 h (Figure 2) (*p* < 0.05). The liposomes showed an initial burst release of about 25.74 ± 5.61%, 26.43 ± 3.81%, 20.65 ± 1.58%, and 22.82 ± 4.71% of GLP from GLP-CL, GLP-DL1, GLP-DL2, and GLP-DL3, respectively, after 2 h, followed by a more gradual release for the following 24 h. The initial burst release might be due to the desorption of GLP from the liposomal surfaces.

The percentage release of the loaded GLP from GLP-CL, GLP-DL1, GLP-DL2, and GLP-DL3 was 58.02 ± 2.08%, 76.28 ± 4.03%, 69.58 ± 2.81%, and 73.65 ± 5.62%, respectively, after 24 h. There was a difference in the release profiles of the CL and DL formulations. The delayed release of the CL formulation drug may be attributable to the GLP needing more time to release from the bilayers of liposomes. While the increased GLP release from DL might be attributed to the fact that the DL can change its shape to pass through smaller holes, this behavior is due to the high vesicle deformability of the membrane in the presence of edge activators [23]. Previous studies showed the same results when comparing the drug’s release from conventional liposomes with deformable ones [43,44].

Moreover, after coating with CS, all formulations displayed a slower release pattern than the uncoated liposomes, which confirmed the coated formation. Additionally, burst release was reduced, which could be clarified by the occurrence of CS on the liposomal surface, which was revealed in the EE% of CS-coated liposomes. The main mechanism described for the decrease in GLP release from CS-coated liposomes is dependent on two points that affect the release rate: the lipid bilayer and the coating layer. These results were consistent with those of a previous report studying the effect of coating liposomes with CS on the drug release pattern [44,45]. The existence of the CS layer delayed the drug diffusion from the phospholipid bilayer into the medium. The liposomes or CS-coated liposomes containing Tween 80 had the highest EE% and higher release rates than the other formulations. All of the investigated formulations behaved as controlled release carriers for at least 24 h.

The in vitro drug release profiles were evaluated in different mathematical models, such as the zero-order, first-order, and Higuchi kinetics models, as shown in Table 3. It was achieved that the Higuchi square root model revealed a higher degree of correlation coefficient (R^2^) than the other models in the case of uncoated liposomes. Hence, the drug release profile of GLP follows the diffusion mechanism. Furthermore, the release data of coated liposomes were fitted using Korsmeyer and Peppas’s well-known empirical equation. As shown, the n (release exponent) value is used to characterize different release mechanisms. It was found that n or the diffusion exponent was 0.5 < *n* < 1, which implies that the drug release from the system follows a non-Fickian transport pattern. In the non-Fickian pattern, the mechanism of drug release is governed by diffusion and swelling. The swelling of the CS coating occurs slowly, and the diffusion process simultaneously causes time-dependent anomalous effects.

### 3.3. Stability of the Liposomes

For safe and efficient use of liposomes, it is essential to establish their stability, which has considerable effects on liposome storage and further applications. A stability study was carried out for short time (28 days) for GLP-loaded CL, DL1, DL2, and DL3 formulations and CS-coated ones. The changes in particle size and PDI are shown in Figure 3. As shown in Figure 3, there were slight changes in the average particle size from the initial values. However, most of the PDI values found throughout the study weeks were less than or equal to 0.5, indicating that uniform vesicles were formed upon storage. Therefore, the prepared formulations showed a high stability during the storage period at 4 °C.

### 3.4. Evaluation of GLP-Loaded Liposome- and CS-Coated Liposomes-Based Transdermal Films

Thus the obtained film was formulated with GLP-loaded coated and uncoated liposomes with two goals, the controlled transdermal delivery and the highest percutaneous of GLP. In consideration of the treatment of hyperglycemia, GLP-loaded film should be of fixed dimensions with content uniformity to give sustained release. The results found that physicochemical characteristics, such as thickness, weight variation, moisture uptake, drug content, and folding endurance, exhibited a slight variation among the obtained films. These parameters evaluated for films are listed in Table 4. The prepared films were smooth in appearance and uniform, without visible cracks. The thicknesses of all films varied from 0.157 ± 0.021 to 0.250 ± 0.017 mm. As shown in Table 4, the weight variation was in the range of 30.96 ± 6.14 to 55.14 ± 3.29 mg. The films produced were found to be satisfactory in terms of their weight variation. The percentage moisture uptake of the tested transdermal films ranged from 5.32 ± 0.69 to 16.81 ± 3.21. The prepared transdermal films showed low percentages of moisture uptake, which are important to ensure the stability of the films and protect them from microbial contamination and bulkiness [30]. As shown in Table 4, the moisture uptake was higher in transdermal films containing chitosan, as it is hydrophilic; a previous study also achieved this result [42]. The folding endurance was found to be <100 with GLP-CL-TDF, GLP-CS-CL-TDF, GLP-DL2-TDF, and GLP-CS-DL2-TDF, which indicated lower strength. However, GLP-DL1-TDF and GLP-DL3-TDF and their CS coating showed a >100 folding endurance, ensuring their acceptable strength, elasticity, and integrity with general skin folding after application. This performance may be attributed to the presence of Tween 80 in GLP-DL1-TDF and GLP-DL3-TDF and their CS coating, which produced a high deformability with a lower DI according to the deformability study. These findings are in agreement with other obtained observations [46]. Uniform drug content was reported for the prepared transdermal films ranging from 97.54 ± 1.89% to 107.40 ± 1.55%. This shows a homogeneous mixture of the drug in the polymer matrix used and the process employed to prepare the films could afford uniform drug content and minimum variability.

### 3.5. Ex Vivo Skin Permeation Studies

To confirm the performance of the sustained release of GLP after transdermal application and observe its in vivo behavior, an ex vivo skin permeation experiment was employed. The experiment was carried out in vertical Franz’s diffusion cells. The permeation results of GLP from the selected different films through excised male Wistar rat skin are presented in Figure 4 and Table 5. The transdermal films containing GLP-DL1-TDF, GLP-CS-DL1-TDF, GLP-DL3-TDF, and GLP-CS-DL3-TDF displayed significantly different permeation patterns through a rat skin membrane. GLP permeation was higher through coated liposomes as compared to non-coated liposomes.

After 24 h, the cumulative amounts of GLP that permeated from GLP-DL1-TDF and GLP-DL3-TDF were 40.645 ± 1.139 and 45.834 ± 3.208 µg/cm^2^, respectively. On the other hand, in CS-coated liposomes, GLP-CS-DL1-TDF and GLP-CS-DL3-TDF delivered 85.117 ± 5.004 and 57.273 ± 2.358 µg/cm^2^ of GLP, respectively, after 24 h. These data showed a sustained release behavior during the 24 h release study. The initial drug release usually occurs due to the fast release of the drug entrapped near the surface of films for which the film drying process was employed. The GLP entrapped deeper in the core matrix of the film takes a longer diffusion path to reach the surface. It has been stated that the use of HPMC in the preparation of transdermal film might be responsible for controlling the release of the drug [47]. The addition of HPMC may increase the viscosity of the reservoir solution and, hence, decrease the permeation rate of GLP [47].

Moreover, the amounts of GLP that permeated were also influenced by the type of edge activators contained in the DL. Moreover, GLP-DL1-TDF that contained Tween 80 resulted in a slightly lower permeation than the one consisting of sodium deoxycholate (GLP-DL3-TDF). This may be attributed to the surfactant’s hydrophilicity, which determined the diffusion rate through the hydrophilic matrix of the transdermal film. The existence of sodium deoxycholate, a hydrophilic surfactant with an HLB value equal to 16, in GLP-DL3-TDF, produced liposome vesicles with a total hydrophilicity higher than GLP-DL1-TDF that consisted of Tween 80 and HLB equal to 15. The presence of Tween 80, as the only surfactant utilized in GLP-DL1-TDF, led to a liposomal vesicle with a relatively lower hydrophilicity. This might interfere with the vesicle mobility within the polymer matrix, allowing a lower GLP delivery through the skin [48]. Therefore, the main mechanism for the increased penetration of GLP-DL3-TDF is the high affinity of phospholipid for biological membranes and increasing hydration by contact with SC [48]. Thus, the packed lipid structure of the SC may disrupt temporarily, allowing the drug to permeate through more readily [48].

Furthermore, the cationic nature of CS-coated liposomes also plays a significant role in improving the rate of GLP permeation. High amounts of GLP were detected for GLP-CS-DL1-TDF and GLP-CS-DL3-TDF. The mechanism for the enhancement of the GLP permeation across the rat skin membrane could be attributed to an electrostatic attraction that developed between the negatively charged skin surface and the positive charge of CS-coated liposomes. The positively charged liposomes have the potential to disrupt the negatively charged tight junctions in the skin. The bioadhesion force of CS could improve the vesicle’s contact period with the skin, causing higher penetration. Many studies have concluded that the vesicles’ charge has a significant impact on the permeation of the drugs through the skin. Positively charged vesicles improve the drug permeation more than negatively charged ones [48]. Thus, the liposomal surface charge represents an additional factor affecting GLP permeation into the skin. CS-coated DL with anionic surfactant (GLP-CS-DL3-TDF) exhibited a more sustained skin permeation of GLP compared to the other with a neutral one (GLP-CS-DL1-TDF). Considering the possible electrostatic interactions that can be triggered between CS and sodium deoxycholate, the total positive charge of the vesicles was decreased. As a result, the skin’s attraction to GLP-CS-DL3-TDF was reduced; hence, a lower GLP permeation was observed [48]. However, more significant amounts of GLP were permeated via GLP-CS-DL3-TDF, revealing the positive influence of the sole surfactant on the skin permeation of GLP.

### 3.6. Permeation Parameters

The skin permeation parameters for GLP-DL1-TDF, GLP-CS-DL1-TDF, GLP-DL3-TDF, and GLP-CS-DL3-TDF directly applied were calculated and are listed in Table 5. The coefficient of determination (R^2^) of the regression lines could be used to assess the stability of drug permeation. The better the stability of the steady-state flux (J) is, the closer the value of *R^2^* is to 1. According to Table 5, the R^2^ values of GLP-CS-DL1-TDF and GLP-CS-DL3-TDF directly applied were both higher than 0.98, showing great stabilities of steady-state flux (J) of 8.436 ± 0.915 and 5.421 ± 0.829 μg/cm^2^/h, respectively. Furthermore, GLP-CS-DL1-TDF and GLP-CS-DL3-TDF showed shorter lag times of 2.165 ± 0.136 and 3.778 ± 0.341 h, respectively. The transdermal *J* of GLP from the GLP-DL1-TDF and GLP-DL3-TDF were 2.725 ± 0.071 and 4.322 ± 0.798 μg/cm^2^/h, respectively. The lag time was 7.092 ± 1.233 and 4.043 ± 0.838 h for GLP-DL1-TDF and GLP-DL3-TDF, respectively, which was larger than that of the other coated DL transdermal film. Due to its higher release rate, the lag time was shorter as well. However, the difference in the cumulative permeation parameters of films suggested the critical role of the formulation composition, resulting in diverse skin permeation profiles. The increased positivity of the formulation could increase the skin permeation of GLP from the corresponding transdermal film. This might be clarified by the fact that CS coating might facilitate interaction with the stratum corneum (the main barrier in drug skin transport). Since GLP-CS-DL1-TDF and GLP-CS-DL3-TDF have proved their transdermal ability in ex vivo studies, in vivo experiments (pharmacokinetic and blood glucose level tests) were conducted to further investigate their in vivo bioavailability and hypoglycemic consequences.

### 3.7. Scanning Electron Microscopy (SEM) of Transdermal Film

The SEM of the selected transdermal films (Figure 5) showed that GLP-CS-DL1 and GLP-CS-DL3 were dispersed in the matrix of the film with some projections on the surface of the film. No difference was observed in the SEM of different formulations.

### 3.8. Pharmacokinetic Studies

The plasma concentrations of GLP formulations after oral and transdermal administration are shown in Figure 6 and the pharmacokinetic parameters are presented in Table 6. The plasma GLP concentration reached C_max_ (13.63 ± 1.18 μg/mL) at a T_max_ of 2.00 ± 0.49 h after oral administration, followed by a rapid decrease in GLP at 24 h and 48 h. In the case of the transdermal film containing GLP (GLP-TDF), the values of C_max_ and T_max_ were 2.42 ± 0.47 μg/mL and 12±00 h, respectively. These values were significantly different compared to those found for the oral route of GLP (*p* < 0.05). While transdermal film containing GLP-CS-DL1-TDF showed a C_max_ of 7.87 ± 1.06 μg/mL and a T_max_ of 12.00 h, GLP-CS-DL3-TDF showed a C_max_ of 4.82 ± 1.03 μg/mL and T_max_ of 12.00 h. These values were significantly different compared to those found for the oral route of GLP (*p* < 0.05). Measurable concentrations of GLP were detected within an hour after the application of the transdermal film and a relatively low C_max_ and prolonged T_max_ (12 h) were observed over 48 h. This behavior of transdermal film related to oral administration could be attributed to the effective barrier of the skin. This could be also due to the quick absorption of drugs via the oral route, with drugs administered through the transdermal route permeating slowly with continuous absorption. Furthermore, the biological half-life (t_1/2_) of GLP after transdermal application was prolonged to about 7.96 ± 0.23–9.73 ± 0.47 h compared to 2.25±0.61 h after oral administration (*p* < 0.05).

The pharmacokinetic profile of GLP when delivered as a transdermal film showed a prolonged release behavior. During the first 12 h, the transdermal film delivered quite a lot of its drug content, followed by an extended release of the drug in the plasma until 48 h. This performance was supported by high t_1/2_ values and increased mean residential time (MRT) values. The MRT of GLP was also prolonged from 4.96 ± 0.55 h (oral) to 27.09 ± 1.08 h (GLP-CS-DL1-TDF). Therefore, the transdermal film could accomplish sustained release and preserve the drug in the body for a longer period. The values of AUC of the transdermal films (GLP-CS-DL1-TDF and GLP-CS-DL3-TDF) were significantly high in comparison with those achieved through oral administration. This effect was attributed to the maintenance of the concentration of GLP for an extended time from the transdermal films. The detected high values of AUC from the transdermal delivery system compared to the oral route indicate the increased bioavailability of GLP in rats. This could be due to the slow and continuous delivery of GLP to the systemic circulation, which may lead to daily glycemic control being achieved in diabetic subjects.

### 3.9. Blood Glucose Levels

The observation of the blood glucose level was considered to estimate the in vivo performance of the hypoglycemic agent, GLP. The change in the blood glucose level–time profile of the five groups is displayed in Table 7. The normal control group (untreated) did not demonstrate significant changes in the blood glucose level during the experiment. The recorded maximum reduction in glucose levels following the administration of oral GLP (138 ± 11.36) was obtained up to 8 h. The hypoglycemic effect of oral GLP was decreased after 8 h, which could be attributed to the short biologic half-life of GLP. Moreover, the blood glucose level after the GLP oral administration was increased from 8 to 48 h, indicating the inability to control the blood glucose level for an extended time. In addition, the results showed that GLP-CS-DL1-TDF or GLP-CS-DL3-TDF provided a reduction in glucose levels of 138 ± 15.53 and 173 ± 9.54, respectively, for 24 h.

The glucose level reduction at 8 h was high with oral administration compared to all other formulations. Moreover, at 48 h, the hypoglycemic effect declined and the blood glucose levels were increased. For GLP-TDF, the decrease in the blood glucose level was monitored for up to 8 h (Table 7). The transdermal delivery of GLP-CS-DL1-TDF or GLP-CS-DL3-TDF could effectively maintain the blood glucose levels in the normal range after 24 h. The data showed that GLP-CS-DL1-TDF can produce a hypoglycemic effect for a prolonged period (up to 24 h) compared to oral GLP. As observed, the severe hypoglycemia induced by oral GLP can be overcome by transdermal systems.

## 4. Conclusions

Different types of DLs were successfully prepared using the thin-film hydration method and the prepared DLs were coated with CS. All the DLs and CS-coated DLs were investigated in terms of their physicochemical parameters and in vitro release. All the liposomes investigated were in the nanometer range with homogenous size and distribution (PDI ≤ 0.5) and sufficient high zeta potential values to provide good stability for the vesicles. Moreover, DLs showed superiority in obtaining high entrapment efficiency over the conventional ones. Because of DLs deformability, the release of loaded GLP through the membrane in the in vitro study increased from the DLs compared to the conventional liposomes. On the other hand, an in vitro release study confirmed that the DLs could reach higher cumulative release amounts of GLP compared to the CS-coated DLs. In addition, the transdermal films containing DLs and CS-coated DLs were prepared with efficient physiochemical properties. An ex vivo skin permeation study demonstrated the sustained release of GLP from the transdermal films, which might allow a more stable plasma level of GLP to be reached over time. CS-coated DLs, such as GLP-CS-DL1 or GLP-CS-DL3, showed a high amount of permeated GLP. GLP-CS-DL1 showed comparable pharmacokinetic parameters to those of oral GLP, but with a much lower C_max_ and higher AUC that could improve the therapeutic efficiency of GLP. Moreover, a higher hypoglycemic effect was obtained after the topical application of the selected formulation of GLP-CS-DL1. Chitosan-coated DL-based transdermal films may prove to be a suitable nano-carrier for percutaneous penetration of GLP. It improves its transport across the skin’s stratum corneum, thus improving its bioavailability.

## Figures and Tables

**Figure 1 pharmaceutics-14-00826-f001:**
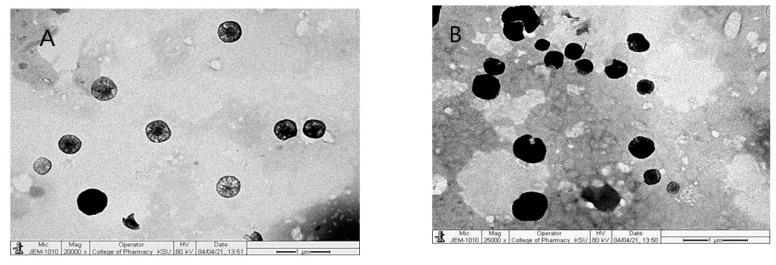
Transmission electron microscope (TEM) images of GLP-CS-DL1 (**A**) and GLP-CS-DL3 (**B**).

**Figure 2 pharmaceutics-14-00826-f002:**
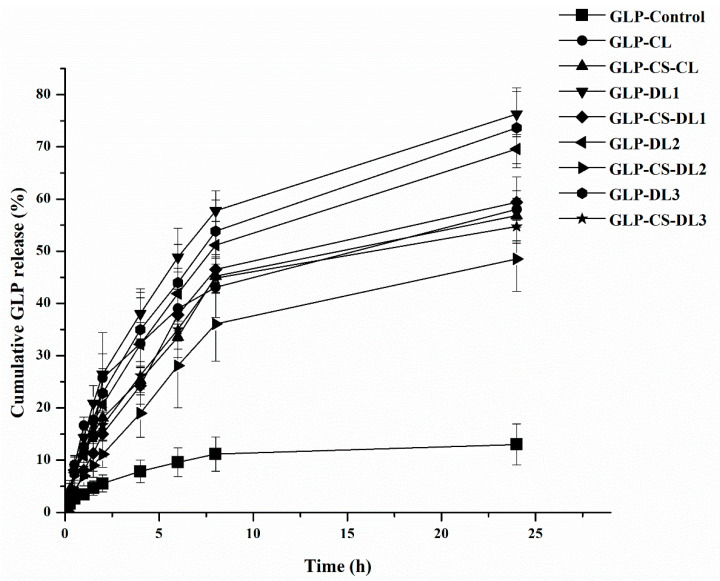
In vitro release profiles of GLP-loaded CL-, DL-, and CS-coated CL/DL in PBS solution (pH 7.4) containing 0.5% Tween 80 (mean ± SD, *n* = 3).

**Figure 3 pharmaceutics-14-00826-f003:**
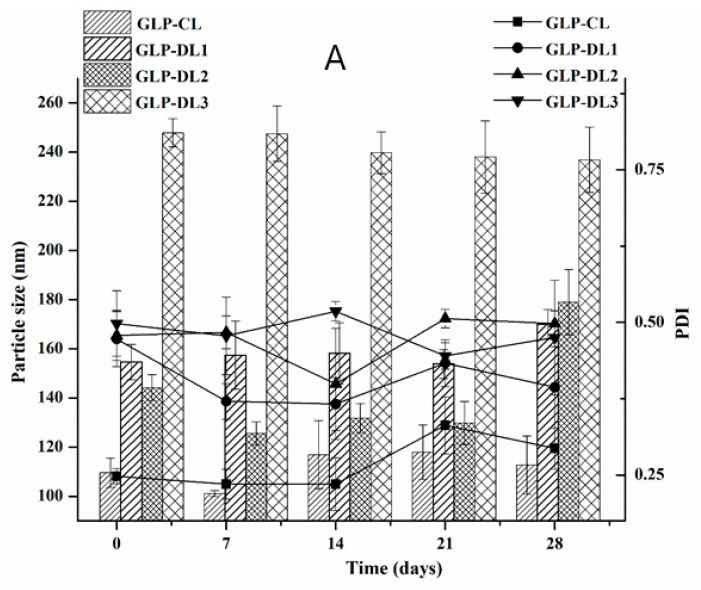
Stability of GLP-loaded CL-, DL- (**A**), and CS-coated CL/ DL (**B**); particle size and PDI (*n* = 3, mean ± SD).

**Figure 4 pharmaceutics-14-00826-f004:**
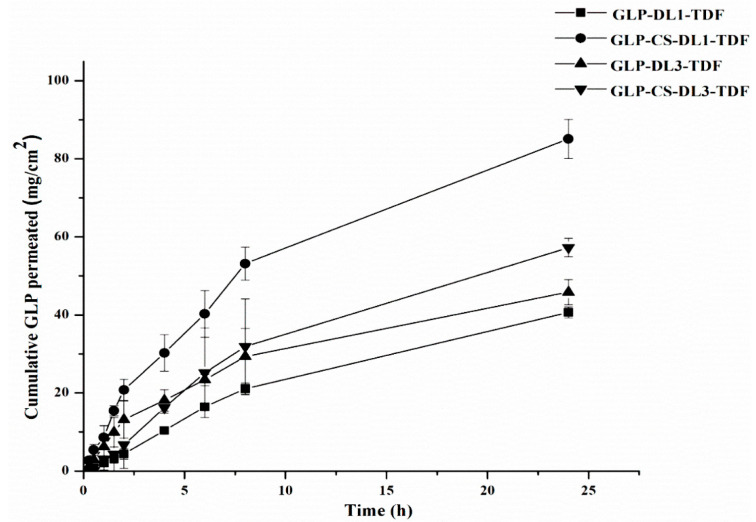
Ex vivo skin permeation profiles of GLP-DL1-TDF, GLP-CS-DL1-TDF, GLP-DL3-TDF, and GLP-CS-DL3-TDF (mean ± SD, *n* = 3).

**Figure 5 pharmaceutics-14-00826-f005:**
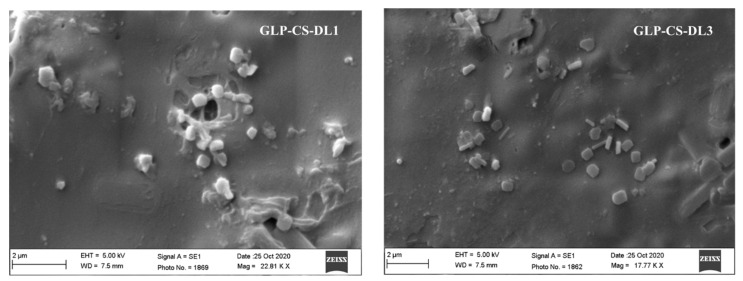
SEM of transdermal films of GLP-CS-DL1 and GLP-CS-DL3.

**Figure 6 pharmaceutics-14-00826-f006:**
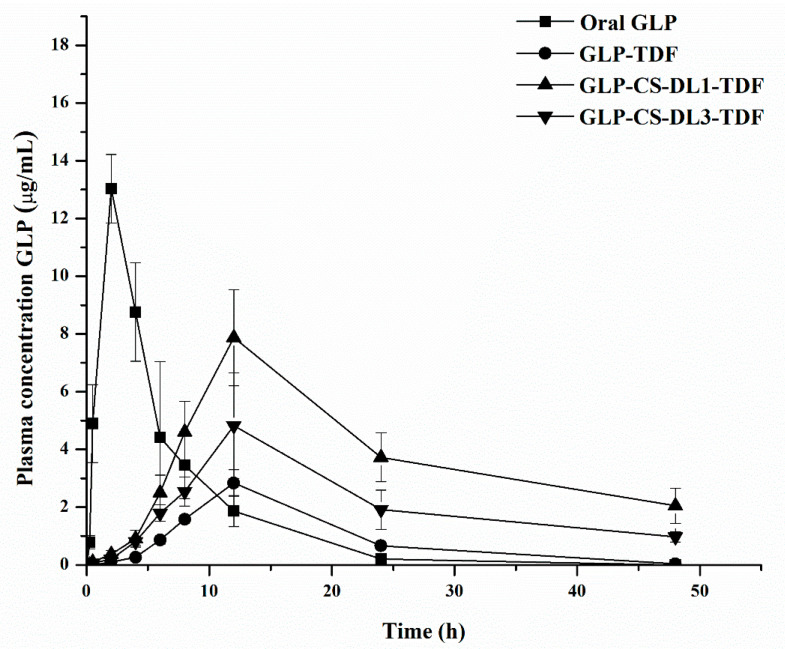
Plasma concentration–time profile of GLP after oral and transdermal film treatment in rats. The results are the mean ± SD, *n* = 5.

**Table 1 pharmaceutics-14-00826-t001:** The composition of different types of CL-, DL-, and CS-coated liposomes containing GLP.

Codes/Ingredients (Molar Ratio)	Lipoid S100	Cholesterol	Tween 80	Na-Deoxycholate	Chitosan (% *w*/*v*)	GLP (% *w*/*v*)
GLP-CL GLP-CS-CL	10.0 10.0	3.0 3.0	0.0 0.0	0.0 0.0	0.0 0.1	0.5 0.5
GLP-DL1 GLP-CS-DL1	9.0 9.0	3.0 3.0	1.0 1.0	0.0 0.0	0.0 0.1	0.5 0.5
GLP-DL2 GLP-CS-DL2	9.0 9.0	3.0 3.0	0.0 0.0	1.0 1.0	0.0 0.1	0.5 0.5
GLP-DL3 GLP-CS-DL3	9.0 9.0	3.0 3.0	0.5 0.5	0.5 0.5	0.0 0.1	0.5 0.5

**Table 2 pharmaceutics-14-00826-t002:** Characterization of the formulated CL-, DL-, and CS-coated liposomes containing GLP.

Formulation Codes	Particle Size (nm ± SD)	PDI (±SD)	Zeta Potential (mV ± SD)	EE% (mean ± SD)
GLP-CL	109.57 ± 5.94	0.248 ± 0.01	−25.71 ± 5.90	79.48 ± 0.55
GLP-CS-CL	178.03 ± 8.69	0.389 ± 0.03	14.26 ± 1.22	64.61 ± 0.5
GLP-DL1	154.92 ±15.82	0.472 ± 0.04	−29.80 ± 3.36	98.65 ± 0.01
GLP-CS-DL1	182.90 ± 1.67	0.423 ± 0.03	16.10 ± 1.05	88.59 ± 0.37
GLP-DL2	144.16 ± 5.20	0.478 ± 0.04	−27.36 ± 2.24	87.32 ± 0.46
GLP-CS-DL2	157.50 ± 2.04	0.444 ± 0.01	17.43 ± 1.17	51.91 ± 1.25
GLP-DL3	247.80 ± 5.75	0.498 ± 0.05	−28.57 ± 5.41	93.25 ± 0.12
GLP-CS-DL3	255.40 ± 6.36	0.382 ± 0.12	24.43 ± 2.83	54.57 ± 1.73

**Table 3 pharmaceutics-14-00826-t003:** In vitro release kinetics models of GLP-loaded CL-, DL-, and CS-coated CL/DL in PBS solution (pH 7.4) containing 0.5% Tween 80.

Formulation Codes	Correlation Coefficient (R^2^)				
	Zero-Order	First-Order	Higuchi’s Model	Korsmeyer–Peppas Model	
				R²	N
GLP-CL	0.8466	0.9099	0.9665	0.9264	0.6282
GLP-CS-CL	0.8891	0.9307	0.9789	0.9884	0.5737
GLP-DL1	0.8751	0.9544	0.9759	0.9568	0.7405
GLP-CS-DL1	0.8862	0.9294	0.9701	0.9754	0.8440
GLP-DL2	0.8973	0.9575	0.9832	0.9813	0.6691
GLP-CS-DL2	0.9090	0.9399	0.9798	0.9873	0.7287
GLP-DL3	0.8955	0.9631	0.9833	0.9776	0.6977
GLP-CS-DL3	0.8731	0.9136	0.9723	0.9827	0.6022

**Table 4 pharmaceutics-14-00826-t004:** Evaluation of GLP-loaded CL-, DL-, and CS-coated liposomes-based transdermal films (TDF).

Formulations Codes	Thickness (mm) ± SD	Weight Variation (mg) ± SD	Moisture Uptake(%) ± SD	Folding Endurance
GLP-CL-TDF	0.202 ± 0.017	41.48 ± 2.24	12.71 ± 2.98	<100
GLP-CS-CL-TDF	0.168 ± 0.032	36.74 ± 8.14	16.11 ± 2.96	<100
GLP-DL1-TDF	0.250 ± 0.017	55.14 ± 3.29	13.02 ± 1.17	>100
GLP-CS-DL1-TDF	0.184 ± 0.009	43.58 ± 1.11	14.19 ± 3.27	>100
GLP-DL2-TDF	0.232 ± 0.010	51.06 ± 2.32	13.68 ± 1.26	<100
GLP-CS-DL2-TDF	0.168 ± 0.035	36.74 ± 8.63	16.81 ± 3.21	<100
GLP-DL3-TDF	0.200 ± 0.032	45.84 ± 10.25	5.32 ± 0.69	>100
GLP-CS-DL3-TDF	0.157 ± 0.021	30.96 ± 6.14	16.10 ± 4.72	>100

**Table 5 pharmaceutics-14-00826-t005:** Cumulative drug permeated and permeation parameters of GLP from different transdermal films (*n* = 3, mean ± SD).

Formulations Code	Cumulative Drug Permeated After 24 h	Flux, J (µg/cm^2^/h)	Permeability Coefficient Kpx10^−4^ (cm h^−1^)	Lag Time (h)	R^2^
GLP-DL1-TDF	40.645 ± 1.392	2.725 ± 0.071	169.497 ± 3.325	7.092 ± 1.233	0.984 ± 0.013
GLP-CS-DL1-TDF	85.117 ± 5.008	8.436 ± 0.915	281.605 ± 13.524	2.165 ± 0.136	0.986 ± 0.012
GLP-DL3-TDF	45.834 ± 3.208	4.322 ± 0.798	204.825 ± 5.792	4.043 ± 0.838	0.978 ± 0.027
GLP-CS-DL3-TDF	57.273 ± 2.358	5.421 ± 0.829	213.402 ± 8.221	3.778 ± 0.341	0.988 ± 0.007

**Table 6 pharmaceutics-14-00826-t006:** Pharmacokinetic parameters of GLP after oral and transdermal administration in rats (*n* = 5, mean ± SD).

Parameters	Oral GLP	GLP-TDF	GLP-CS-DL1-TDF	GLP-CS-DL3-TDF
Cmax (µg/mL)	13.63 ± 1.18	2.42 ± 0.47	7.87 ± 1.06	4.82 ± 1.03
Tmax (h)	2.00 ± 0.49	12.00 ± 00	12.00 ± 0.00	12.00 ± 0.00
AUC (0→24) (µg-h/mL)	70.58 ± 7.26	43.02 ± 4.51	164.12 ± 23.12	96.22 ± 11.78
AUC (0→∞) (µg-h/mL)	70.57 ± 7.27	47.71 ± 5.21	189.07 ± 30.6	109.24 ± 14.97
t1/2 (h)	2.25 ± 0.61	7.96 ± 0.23	9.73 ± 0.47	8.34 ± 0.68
MRT (h)	4.96 ± 0.55	24.44 ± 0.38	27.09 ± 1.08	23.97 ± 2.16

**Table 7 pharmaceutics-14-00826-t007:** Effect on blood glucose levels in STZ-induced diabetic rats (mean ± SD).

Time (h)	Normal Control	Oral GLP	GLP-TDF	GLP-CS-DL1-TDF	GLP-CS-DL3-TDF
0	109 ± 8.18	335 ± 20.67	356 ± 14.11	354 ± 17.15	348 ± 16.50
0.5	119 ± 13.27	349 ± 21.81	356 ± 26.37	357 ± 22.91	368 ± 26.76
1	136 ± 9.62	353 ± 23.01	338 ± 21.73	354 ± 21.78	354 ± 22.56
2	113 ± 15.11	299 ± 21.89	331 ± 9.75	293 ± 22.17	303 ± 25.11
4	121 ± 5.35	216 ± 17.67	310 ± 19.08	252 ± 19.47	268 ± 23.52
6	115 ± 11.95	186 ± 15.95	305 ± 21.84	228 ± 15.56	205 ± 18.68
8	108 ± 17.53	138 ± 11.36	274 ± 18.74	163 ± 10.81	199 ± 15.17
24	128 ± 7.03	150 ± 10.61	319 ± 22.14	138 ± 15.53	173 ± 9.54
48	137 ± 11.02	227 ± 7.07	344 ± 24.54	236 ± 17.89	331 ± 8.51

## Data Availability

The data presented in this study are available on request from the corresponding author.

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
