# Peer review of "Transdermal Glipizide Delivery System Based on Chitosan-Coated Deformable Liposomes: Development, Ex Vivo, and In Vivo Studies"

_pharmaceutics, 2022, doi:10.3390/pharmaceutics14040826_

Round 1

Reviewer 1 Report

Te authors have addressed my comments made from the previous submission of the manuscript.

Author Response

Dear reviewer 1

The authors would like to thank you a lot for your efforts and valuable comments.

Reviewer 2 Report

I still feel this manuscript is not suitable for publication as the developed delivery system showed no improvement for the glucose control and there is no other experiments regarding the DM treatment.

Author Response

Dear reviewer 2

The authors would like to thank you a lot for your efforts and your valuable time for revising the manuscript. We have made a lot of corrections in our study to be clear and understandable with more explanations and more references that were added to reflect the interesting and application of the work.

Reviewer 3 Report

This report (ID: pharmaceutics-1636628) entitled “Transdermal Glipizide delivery system based on chitosan-coated deformable liposomes: Development, ex vivo and in vivo studies“ by research group El Sayeh Abou El Ela.  This article is interesting and can contribute to the scientific literature. However, the authors must address formal flaws and strengthen the presentation and clarity. The reviewer suggests minor. revision is needed before publication in a peer-reviewed journal. If authors improve the current text of the manuscript and improve illustrations of figures, it can be recommended for publication. Strengths and weaknesses of the article, some specific comments on the manuscript:

1 ORIGINALITY:

State the novelty and practical applications.

  1. INTRODUCTION:

The authors have to inform if this article has novelty, is sufficiently innovative or interesting,

  1. METHODOLOGY:

*The authors need to explain methodologies accurately and clearly explain how the data was collected?

  1. RESULTS AND DISCUSSION:

*Cost-effectiveness, advantages of method and materials, mechanism of action is missing in the section.

*The suggestion is to improve flow, English. Overall presentation is descriptive and challenging to follow, improve readability.

*Representation of results and discussion can be further enhanced and focus on scientific aspects.

*Specify limitations of methods and materials at the end of the section.

  1. CONCLUSIONS:

*Ensure conclusions reflect research work resulting in an improvement.

  1. REFERENCES:

*Make sure cited references reflect the leading publications based on the research work.

  1. This paper needs to make significant revisions to improve the presentation and be published in a peer-reviewed scientific journal. Improve readability; it is important to ease with reading to help the reader to understand the text.
  2. Additional comments to help author, kindly address the following comments:
  3. To the best of my knowledge, the manuscript must be technically and scientifically correct. The manuscript is technically weak, improving its current form.
  4. The manuscript needs to comply with standard and journal format.

Author Response

Dear reviewer 3

The authors would like to thank you for your valuable and appreciated comments that added a value to the work.  We have tried to make a response for all your comments. You can find the them in the file attached.

This manuscript is a resubmission of an earlier submission. The following is a list of the peer review reports and author responses from that submission.

Round 1

Reviewer 1 Report

The present manuscript deals with the development of transdermal liposomal formulation for glipizide and their characterization in vitro and in vivo.

Some revisions are required:

Introduction:

Line 55-56 One study performed on liposomal glipizide does not solve any issue related to the patient’s incompliance with this drug. Please revise.

Line 59 Reference [7] is not correct. Please revise.

Line 82 polysorbate 80 is not a polymer. Please revise.

Materials: Please specify the molecular weight and degree of acetylation of chitosan used.

Please specify the molecular weight and the type of HPMC and carboxymethyl cellulose used.

Line 142-147 It is not clear how chitosomes were prepared, which is the concentration of chitosan solution used and which is the final concentration of chitosan and, especially, liposomes in the final formulation.

Line 174 How the size of the vesicles was determined?

Line 204-205 How “vesicle size, PDI or zeta potetintial” can be assessed macroscopically?

Line 243 and line 167 Why two different references ([16] and [22]) were cited for the HPLC method used for glipizide determination in two different paragraphs?

Line 292-293 How transdermal films were applied on rats for in vivo studies?

Paragraph 3.1.3 The concepts expressed here for the deformability of liposomes seem not to be correct. According to the equation in method line 177, liposomes should be more deformable when the deformability index is high. Liposomes are more deformable if they are simpler extruded.

Line 650-651 the mechanical properties of the film cannot be assessed and defined efficient if only tested manually.

Reviewer 2 Report

This report (ID: pharmaceutics-1609698) entitled “Transdermal Glipizide delivery system based on chitosan-coated deformable liposomes: Development, and ex vivo and in vivo studies “by research group Amal El Sayeh Abou El 5 Ela. This article is interesting and can be an important contributor to the scientific literature. However, the authors have to address formal flaws and strengthen the presentation and clarity. The reviewer suggests major revision is needed before publication in a peer-reviewed journal. If authors improve the current text of the manuscript and improve illustrations of figures, it can be recommended for publication. Strengths and weaknesses of the article, some specific comments on the manuscript:

  1. Originality

The authors have to inform if this article has novelty, is sufficiently innovative or interesting makes it commendable to publish.

  1. Abstract

The suggestion to authors is, to revise the abstract to reflect the need or summarize the problem, mention used methods, obtained results, and share some conclusions. Suggestion to include novelty illustration in graphical abstract.

  1. Methodology:

*The authors need to explain methodologies accurately with clarity explain, mention how the data was collected?

*There is a need for sufficient data sets and experiments, which can help other researchers reproduce.

  1. Results and discussion:

*Make sure it is strong, does it show the scientific clarity, mechanisms, and illustration of results.  

*Cost-effectiveness, advantages of method and materials, mechanisms related details are missing in the section.

*The suggestion is to improve flow, English. The overall presentation is descriptive and difficult to follow, improving readability.

*Representation of results and discussion can be further enhanced and focus on scientific aspects.

*Specify limitations of methods and materials at the end of the section.

  1. Conclusion

*Ensure conclusions reflect research work resulting in an improvement.

*Do not duplicate the text of the abstract or results.

*Focus on the scope (such as future implications, sustainability, cost-effectiveness, improvement in performance)

  1. References – make sure cited references reflects the main publications on which the research work is based.

  1. This paper needs to make significant revisions in order to improve the presentation and to be published in a scientific peer-reviewed journal. To improve readability, it is important to ease with reading, to help readers to understand the text.
  2. Additional comments to help author, kindly address the following comments:
  3. To the best of my knowledge, the manuscript must be technically and scientifically correct. The manuscript is technically weak, improving its current form.
  4. The manuscript needs to comply with standard and journal format.

Reviewer 3 Report

In this manuscript, the authors developed several types of liposome-based formulations for controlled release and transdermal delivery of glipizide (GLP). The authors comprehensively evaluated and compared properties of degradable liposomes, chitosan-coated liposomes and their film formulations. Transdermal delivery of GLP for diabetes therapy is not particularly new. Although GLP-CS-DL1 showed comparable pharmacokinetic parameters to those of oral GLP, it seems that the GLP-CS-DL-TDF has little benefits in glucose control in STZ-induced diabetic rats compared to oral GLP. Therefore, this manuscript is not suitable for publication in the journal.

  1. What is the oral administration problem for GLP? It is low cost and have high compliance in patients. More importantly, oral administration of GLP is effective for glucose control.
  2. For the blood glucose levels (Table 7), it would be helpful to use a line graph.